# Usability and acceptability of oral fluid- and blood-based hepatitis C virus self-testing among the general population and men who have sex with men in Malaysia

**Huan-Keat Chan**[1], **Xiaohui Sem**[2], **Elena Ivanova Reipold**[2]*, **Sheela Bai A/P Pannir Selvam**[3], **Narul Aida Salleh**[4], **Abdul Hafiz Bin Mohamad Gani**[5], **Emmanuel Fajardo**[2], **Sonjelle Shilton**[2], **Muhammad Radzi Abu Hassan**[1]

1 Clinical Research Centre, Hospital Sultanah Bahiyah, Alor Setar, Malaysia, 2 FIND, Geneva, Switzerland, 3 Cheras Health Clinic, Kuala Lumpur, Malaysia, 4 Kuala Lumpur Health Clinic, Kuala Lumpur, Malaysia, 5 Mahmoodiah Health Clinic, Johor Bharu, Malaysia

* elena.ivanova@finddx.org

**Data Availability Statement:** The authors confirm that the data supporting the findings of this study

## Abstract

Hepatitis C self-testing (HCVST) is emerging as an additional strategy that could help to expand access to HCV testing. We conducted a study to assess the usability and acceptability of two types of HCVST, oral fluid- and blood-based, among the general population and men who have sex with men (MSM) in Malaysia. An observational study was conducted in three primary care centres in Malaysia. Participants who were layman users performed the oral fluid- and blood-based HCVST sequentially. Usability was assessed by calculating the rate of errors observed, the rate of difficulties faced by participants as well as inter-reader (self-test interpreted by self-tester vs interpreted by trained user) and inter-operator concordances (self-test vs test performed by trained user). The acceptability of HCV self-testing was assessed using an interviewer-administered semi-structured questionnaire. Participants were also required to read contrived test results which included "positive", "negative", and "invalid". There was a total of 200 participants (100 general population, 100 MSM; mean age 33.6 ± 14.0 years). We found a high acceptability of oral fluid- and blood-based HCVST across both general population and MSM. User errors, related to timekeeping and reading within stipulated time, were common. However, the majority of the participants were still able to obtain and interpret results correctly, including that of contrived results, although there was substantial difficulty interpreting weak positive results. The high acceptability of HCVST among the participants did not appreciably change after they had experienced both tests, with 97.0% of all participants indicating they would be willing to use HCVST again and 98.5% of them indicating they would recommend it to people they knew. There was no significant difference between the general population and MSM in these aspects. Our study demonstrates that both oral fluid- and blood-based HCVST are highly acceptable among both the general population and MSM. Both populations also showed comparable ability to conduct the tests and interpret the results. Overall, this study suggests that HCVST could be introduced as an addition to existing HCV testing services in Malaysia.

are available within the article and its supplementary materials.

**Funding:** The Government of Netherlands funded the study but had no role in study design, data collection and analysis, decision to publish, or preparation of the manuscript.

**Competing interests:** The authors have declared that no competing interests exist.

Further studies are needed to establish the optimal positioning of self-testing alongside facility-based testing to expand access to HCV diagnosis in the country.

## Introduction

Hepatitis C virus (HCV) infection remains a major contributor to life-threatening liver diseases worldwide. Approximately 58 million individuals are currently living with hepatitis C [1, 2]. The advent of highly effective direct-acting antivirals (DAAs), coupled with the World Health Organization's (WHO) goal to eliminate HCV as a public health threat by 2030 [3], has prompted global screening and treatment scale-up efforts.

Although globally 9.4 million people with HCV benefited from simplified testing procedures and received DAA-based treatment between 2015 and 2019 [4], the suboptimal uptake of facility-based testing, largely due to the poor accessibility of primary care, concerns around stigma, and competing priorities [5, 6], has limited the expansion of treatment. Adopting cost-effective yet user-friendly innovations, particularly self-testing, may help overcome such challenges and make decentralized HCV testing a success [7].

Self-testing is a process in which an individual collects a specimen, performs a test and interprets the results on their own [8, 9]. HIV self-testing (HIVST) has been used to complement conventional facility-based HIV testing services [10]. It has been shown to substantially enhance HIV testing uptake, especially among men who have sex with men (MSM) [11–14]. It is important to note that with HCVST, as it is an anti-HCV antibody test, a positive result may not indicate active viremia [5]. This differs from HIVST and requires supportive education such that test performers understand how to interpret what a positive HCVST may mean. Given the overlapping risk factors and challenges of HCV and HIV, policy and programme synergies in national responses to these infections have been advocated [15]. Recently, WHO issued a new recommendation to use HCV self-testing (HCVST) as an additional approach to supplement facility-based testing services [16]. It is critical that for countries keen to introduce HCVST into their testing policy, local evidence should be generated for acceptability, values and preferences to inform HCVST implementation and the WHO's recommendations should be adapted accordingly to the national and local context.

Malaysia, an upper middle-income country with a population of 32.7 million, has an anti-HCV prevalence of approximately 1.9% in the general population [17] and 4.6% in MSM [18]. The national response to HCV in Malaysia is mainly driven by the Ministry of Health (MOH), based on principles of improving treatment accessibility and resource optimization. Despite initial successes in the expansion of DAA-based treatment [19, 20], a large majority of those infected remain undiagnosed and the MOH continues to collaborate with FIND, a non-profit diagnostics initiative, to explore options increasing access to HCV screening.

Recent studies of oral fluid-based HCVST conducted in Brazil, China, Egypt, Georgia, Kenya and Vietnam have shown overall high usability and acceptability [21–26], while blood-based HCVST was shown to have high usability in South Africa (in the general population) [27]. In this study, we made the first attempt to determine the usability and acceptability of both oral fluid- and blood-based HCVST among the general population and MSM in Malaysia, including understanding how well participants can interpret HCVST and contrived results.

## Methods

### Study design and setting

This cross-sectional study was undertaken in three public primary care centres located in the State of Johor (Mahmoodiah Health Clinic) and the Federal Territory of Kuala Lumpur (Cheras Health Clinic and Kuala Lumpur Health Clinic). Apart from providing general medical care, these three centres also run special clinics for sexually transmitted diseases (STDs), where approximately 70% of patients are MSM. These centres also provide free HIV treatment for Malaysian citizens. This study (protocol number: NMRR-20-1794-56098) was approved by the Medical Research Ethics Committee of the MOH.

### Study participants and sample size

Participants were recruited via convenience sampling during a 2-month period (18 December 2020 to 21 February 2021). They were all adults (≥18 years of age) who either self-referred to participate in this study or were receiving care at one of the three study sites at the time of being approached. All participants were considered to have an unknown HCV status as they either have never been tested for HCV before or tested negative more than 6 months prior to study enrollment. They also had no experience with either HIVST or HCVST.

Based on expert opinion derived from experience with HIVST and the finding of a recent study comparing PWID and MSM when performing HCVST [21], this study was powered to detect a 20% difference in the usability of HCVST between the general population and MSM (70% versus 50%). The level of significance ($\alpha$) and power (1-$\beta$) was fixed at 5% and 80%, and the minimum sample sizes required for the general population and MSM were therefore 94 individuals each. Assuming a 5% attrition rate, a sample-size target of 100 participants per group was set.

### HCV self-testing kits

All participants performed HCVST with two kits: OraQuick (oral fluid-based test kit; manufactured by OraSure Technologies, United States of America) and First Response (blood-based test kit; manufactured by Premier Medical Corporation, India). OraQuick (sensitivity: 98.1%; specificity: 99.6%) was already prequalified by the WHO for professional use at the time of this study, whereas First Response (sensitivity: 100%; specificity: 100%) was still under the review of the WHO for professional use. Both kits were repackaged and adapted for HCVST by the manufacturers and labelled "for research use only". Instructions for use (IFU) in both Bahasa Malaysia and English were also included in the package. Both tests were active. No results from either test were used to guide any clinical decisions. All participants received a standard HCV test at the end of the study and were linked to treatment if necessary.

### Study procedures

At the study sites, individuals were briefed about the study and screened for eligibility. After providing written consent, study procedures were carried out in a private room. A structured questionnaire was used to record their demographic data, past exposure to HCV risk factors, experience with HCV and HIV testing, and willingness to use HCVST. Participants were then provided with both written (in Bahasa Malaysia or English) and pictorial instructions about HCVST.

To reduce usability bias, every other participant at each study site performed the oral fluid-based test first, while the remainder used the blood-based test first. Participants were advised to avoid consuming food or using oral care products for 15 and 30 minutes, respectively,

before performing the oral fluid-based test. For the oral fluid-based test, participants swabbed their upper and lower gums with the test device, placed it in the provided tube, and read the results within 20 to 40 minutes of taking the test. For the blood-based test, participants collected a drop of blood by pricking a finger with a lancet, transferred the blood onto the test device, added the diluent, and read the results after 15 minutes. Any errors observed during self-testing and result interpretation by the participant were documented by a study team member using a checklist. No assistance was provided unless requested by a participant and after at least 15 minutes of repeated efforts to conduct each testing step unassisted.

To determine inter-reader concordance, a second study team member, who was blinded to the self-reported test results, re-read the results within the stipulated time. Subsequently, to determine inter-operator concordance, a third study team member, who was blinded to both the self-reported and re-read results, performed professional-use oral fluid- and blood-based tests. Next, participants were required to read contrived test results from premade test cassettes which included a mix of "positive", "negative", and "invalid" results (S1 Fig). Finally, a structured interview was conducted to explore the acceptability of HCVST among participants, as well as to collect additional information regarding their awareness of HCV and its treatment.

### Data assessment and analysis

The usability of HCVST was expressed as the proportions of participants who were observed to make mistakes, incorrectly interpret results, experience difficulties or require assistance when performing the tests. The inter-reader and inter-operator concordances of result interpretation represented the degrees (in percentages) to which the test results reported by the participants agreed with those re-read and found in the repeated tests, respectively. The Gwet's AC1 coefficient was also used to measure the inter-reader and inter-operator reliability of result interpretation, given the imbalanced distribution of HCV test results across categories [28]. Acceptability of HCVST among the participants, along with their awareness of hepatitis C and its treatment, were summarized as numbers and percentages. Pearson's chi-square or Fisher's exact test was used to explore associations between two categorical variables, while the Mann–Whitney U test was used to compare means between two groups of participants. Statistical analyses were performed using SPSS Statistics V21.0 (IBM, New York), with the level of significance set at $p<0.05$.

## Results

### Recruitment and characteristics of participants

Of 221 individuals approached, 216 (97.7%) fulfilled the eligibility criteria and 200 (92.6%) consented to participate in this study. The number of participants recruited in the three study sites was 60 (30.0%), 66 (33.0%) and 74 (37.0%) respectively. All participants completed both oral fluid- and blood-based HCVST, as well as the post-testing interviews (Fig 1).

Table 1 shows the participants' self-reported characteristics; their mean age was 33.6 ± 14.0 years; most were male (75.0%) and unmarried (65.0%). MSM were more likely to make >1 visit to clinics annually compared with the general population (80.0% vs. 39.0%, $p < 0.001$). The majority of the MSM participants have been previously tested for HIV (96.0% vs 50%, $p < 0.001$); 59% of them reported a positive HIV test result. Although none of the participants had prior experience with HIV or HCV self-testing, participants from the general population had more experience with performing home-based testing such as pregnancy, glucose monitoring and blood pressure tests compared to MSM (43.0% vs. 16%; $p < 0.001$). Before taking the self-test, 83.0% of the participants across both population groups were aware of self-testing

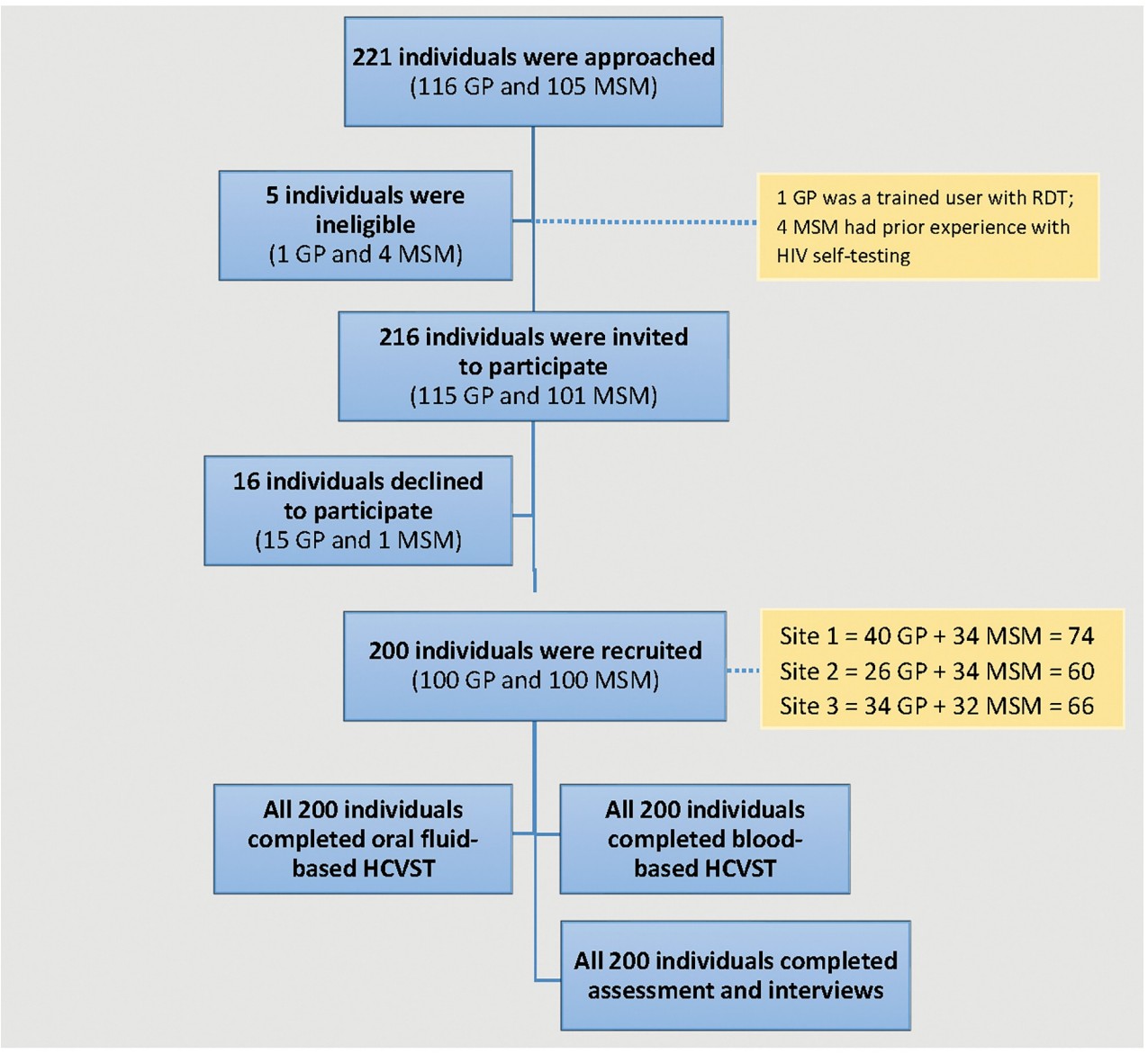

GP, general population; HCVST, hepatitis C self-testing; MSM, men who have sex with men.

**Fig 1. Flowchart of participant enrolment.** GP, general population; HCVST, hepatitis C self-testing; MSM, men who have sex with men.

approaches in general and almost all the participants (98.0%) expressed their willingness to use HCVST if it was available.

## Usability of HCVST

For the oral fluid-based HCVST, 53.0% of the participants completed the self-test procedure without any error. The most common mistakes observed during the pre-testing steps were not practicing proper timekeeping (24.5%) and not reading the test results within the stipulated time (38.5%). Excluding these two mistakes, 87.5% of the participants completed the self-test procedure without any error. In addition, 88.5% of the participants completed the self-test procedure without any difficulty and 95.5% completed the self-test procedure without any

**Table 1. Self-reported characteristics of participants: Overall, general population and men who have sex with men.**

| Characteristics | Overall (n = 200) | Subgroup | | |
|---|---|---|---|---|
| | | GP (n = 100) | MSM (n = 100) | p-value |
| **Age, median (IQR)** | 33.6 (14.0) | 35.8 (19.0) | 29.5 (11.0) | 0.12 [a] |
| **Sex, n (%)** | | | | - |
| Male | 157 (78.5) | 57 (57.0) | 100 (100.0) | |
| Female | 43 (21.5) | 43 (43.0) | - | |
| **Education level, n (%)** | | | | 0.12 [b] |
| Tertiary | 104 (52.0) | 45 (45.0) | 59 (59.0) | |
| Secondary/post-secondary | 90 (45.0) | 51 (51.0) | 39 (39.0) | |
| No formal education/primary | 6 (0.3) | 4 (4.0) | 2 (2.0) | |
| **Work status, n (%)** | | | | 0.018 [c] |
| Employed/self-employed | 163 (81.5) | 75 (75.0) | 88 (88.0) | |
| Unemployed | 37 (18.5) | 25 (25.0) | 12 (12.0) | |
| **Marital status, n (%)** | | | | <0.001 [b] |
| Unmarried | 130 (65.0) | 39 (39.0) | 91 (91.0) | |
| Married or living with a partner | 61 (30.5) | 54 (54.0) | 7 (7.0) | |
| Divorced, separated or widowed | 9 (4.5) | 7 (7.0) | 2 (2.0) | |
| **History of exposure to HCV risk factors, n (%)** | | | | |
| Condomless anal intercourse | 80 (40.0) | - | 80 (80.0) | - |
| Dental procedure | 72 (36.0) | 42 (42.0) | 30 (30.0) | 0.08 [c] |
| Surgical procedure | 38 (19.0) | 23 (23.0) | 15 (15.0) | 0.15 [c] |
| Sharing shaving tool or toothbrush | 14 (7.0) | 5 (5.0) | 9 (9.0) | 0.27 [c] |
| Having a tattoo | 4 (2.0) | 1 (1.0) | 3 (3.0) | 0.62 [b] |
| Injecting non-prescription drugs | 2 (1.0) | 0 (0.0) | 2 (2.0) | 0.50 [b] |
| **Frequency of clinic visit, n (%)** | | | | <0.001 [b] |
| >1 time per year | 119 (59.5) | 39 (39.0) | 80 (80.0) | |
| 1 time per year | 42 (21.0) | 27 (27.0) | 15 (15.0) | |
| Rarely | 33 (16.5) | 29 (29.0) | 4 (4.0) | |
| Never | 6 (3.0) | 5 (5.0) | 1 (1.0) | |
| **Most recent HCV test result, n (%)** | | | | <0.001 [c] |
| Negative | 42 (21.0) | 9 (9.0) | 33 (33.0) | |
| Never tested | 158 (79.0) | 91 (91.0) | 67 (67.0) | |
| **Latest HIV test result, n (%)** | | | | <0.001 [b] |
| Positive | 60 (30.0) | 1 (1.0) | 59 (59.0) | |
| Negative | 86 (43.0) | 49 (49.0) | 37 (37.0) | |
| Never tested | 54 (27.0) | 50 (50.0) | 4 (4.0) | |
| **Aware that at least one type of home-based self-testing approach is available, n (%)** | 166 (83.0) | 87 (87.0) | 79 (79.0) | 0.13 [c] |
| **Has experience with performing any home-based self-testing, n (%)** | 59 (29.5) | 43 (43.0) | 16 (16.0) | <0.001 [c] |
| **Willing to use home-based HCV self-testing if it is made available [d]** | 196 (98.0) | 98 (98.0) | 98 (98.0) | >0.95 [b] |

HCV, hepatitis C virus; GP, general population; HIV, human immunodeficiency virus; IQR, interquartile range; MSM, men who have sex with men.

[a] Mann–Whitney U test

[b] Fisher's exact test

[c] Pearson's chi-square test

[d] Asked before performing HCVST.

**Table 2.** Assessment of mistakes, difficulties and assistance required in oral fluid-based self-testing: Overall, general population and men who have sex with men.

| Step | Overall (n = 200) | Subgroup | | |
|---|---|---|---|---|
| | | GP (n = 100) | MSM (n = 100) | *p*-value |
| **No. of participants who made errors in pre-testing steps, n (%)** | | | | |
| Opening the package | 0 (0.0) | 0 (0.0) | 0 (0.0) | - |
| Reading and using the instructions for use | 0 (0.0) | 0 (0.0) | 0 (0.0) | - |
| Removing the test tube from the test pack | 0 (0.0) | 0 (0.0) | 0 (0.0) | - |
| Removing the cap from the test tube | 0 (0.0) | 0 (0.0) | 0 (0.0) | - |
| Placing the tube into the stand | 3 (1.5) | 0 (0.0) | 3 (3.0) | 0.25 [a] |
| Removing the test device from the test pack | 0 (0.0) | 0 (0.0) | 0 (0.0) | - |
| *No. of participants who made at least one error, n (%)* | 3 (1.5) | 0 (0.0) | 3 (3.0) | 0.25 [a] |
| **No. of participants who made errors in testing steps, n (%)** | | | | |
| Avoidance of touching the flat pad | 12 (6.0) | 7 (7.0) | 5 (5.0) | 0.52 [b] |
| Collecting oral fluid | 14 (7.0) | 4 (4.0) | 10 (10.0) | 0.10 [b] |
| Placing of test device in the tube | 2 (1.0) | 0 (0.0) | 2 (2.0) | 0.50 [a] |
| Timekeeping | 49 (24.5) | 28 (28.0) | 21 (21.0) | 0.25 [b] |
| Reading results within 20–40 minutes after the test | 77 (38.5) | 41 (41.0) | 36 (36.0) | 0.47 [b] |
| *No. of participants who made at least one error, n (%)* | 94 (47.0) | 47 (47.0) | 47 (47.0) | >0.95 [b] |
| *No. of participants who made at least one error (excluding those related to timekeeping and reading results within stipulated time), n (%)* | 25 (12.5) | 11 (11.0) | 14 (14.0) | 0.52 [b] |
| **No. of participants with the following observed difficulty, n (%)** | | | | |
| Opening the package | 2 (1.0) | 2 (2.0) | 0 (0.0) | 0.50 [a] |
| Opening the test tube | 5 (2.5) | 5 (5.0) | 0 (0.0) | 0.06 [a] |
| Sliding the tube into the stand | 13 (6.5) | 4 (4.0) | 9 (9.0) | 0.15 [b] |
| Placing the test device into the tube | 2 (1.0) | 0 (0.0) | 2 (2.0) | 0.50 [a] |
| Reading and interpreting the results | 3 (1.5) | 1 (1.0) | 2 (2.0) | >0.95 [a] |
| *No. of participants with observed difficulties for at least one step, n (%)* | 23 (11.5) | 12 (12.0) | 11 (11.0) | 0.83 [b] |
| **No. of participants requiring assistance, n (%)** | | | | |
| Opening the package | 0 (0.0) | 0 (0.0) | 0 (0.0) | - |
| Opening the test tube | 2 (1.0) | 1 (1.0) | 1 (1.0) | >0.95 [a] |
| Sliding the tube into the stand | 7 (3.5) | 2 (2.0) | 5 (5.0) | 0.45 [a] |
| Placing the test device into the tube | 1 (0.5) | 0 (0.0) | 1 (1.0) | >0.95 [a] |
| Reading the results | 0 (0.0) | 0 (0.0) | 0 (0.0) | - |
| *No. of participants requiring assistance for at least one step, n (%)* | 9 (4.5) | 3 (3.0) | 6 (6.0) | 0.50 [a] |

GP, general population; MSM, men who have sex with men.

[a] Fisher's exact test

[b] Pearson's chi-square test

assistance. The general population and MSM did not differ in any of the above aspects of oral fluid-based HCVST (Table 2).

In terms of the blood-based HCVST, 28.0% of the participants completed the self-test procedure without any error. The most common mistakes observed during the pre-testing steps were not washing hands in warm water and drying them before performing the test (58.0%), not waiting long enough (15 minutes after adding the diluent) to read the test results (44.0%), not choosing a middle or ring finger to prick for the test (43.5%), not practicing correct timekeeping (35.0%), not discarding the first drop and using the second drop of blood for the test (22.0%), and not massaging and warming the finger before the test (20.0%). Despite these errors, 62.5% of the participants completed the self-test procedure

**Table 3. Assessment of mistakes, difficulties and assistance required in blood-based self-testing: Overall, general population and men who have sex with men.**

| Step | Overall (n = 200) | Subgroup | | |
|---|---|---|---|---|
| | | GP (n = 100) | MSM (n = 100) | *p*-value |
| **No. of participants who made errors in pre-testing steps, n (%)** | | | | |
| Opening the package | 0 (0.0) | 0 (0.0) | 0 (0.0) | - |
| Reading and using the instructions for use | 1 (0.5) | 0 (0.0) | 1 (1.0) | >0.95 [a] |
| Removing the test device from the foil pouch | 0 (0.0) | 0 (0.0) | 0 (0.0) | - |
| Washing hands in warm water and drying them | 116 (58.0) | 61 (61.0) | 55 (55.0) | 0.39 [b] |
| Choosing a middle or ring finger | 87 (43.5) | 43 (43.0) | 44 (44.0) | 0.89 [b] |
| Massaging and warming the finger | 40 (20.0) | 20 (20.0) | 20 (20.0) | >0.95 [b] |
| Cleaning fingertip with alcohol pad | 3 (1.5) | 2 (2.0) | 1 (1.0) | >0.95 [a] |
| **No. of participants who made errors in testing steps, n (%)** | | | | |
| Pressing the lancet against the finger to prick skin | 12 (6.0) | 11 (11.0) | 1 (1.0) | 0.003 [b] |
| Wiping away the first drop of blood and rubbing to create a second | 44 (22.0) | 23 (23.0) | 21 (21.0) | 0.73 [b] |
| Using the transfer device to collect the drop of blood | 8 (8.0) | 3 (3.0) | 5 (5.0) | 0.72 [a] |
| Dispensing the whole blood into the round specimen well | 2 (1.0) | 2 (2.0) | 0 (0.0) | 0.50 [a] |
| Applying the plaster | 25 (12.5) | 11 (11.0) | 14 (14.0) | 0.52 [b] |
| Twisting and pulling the cap to open assay diluent, then dispensing two drops of the assay diluent into the specimen well | 14 (7.0) | 4 (4.0) | 10 (10.0) | 0.10 [b] |
| Timekeeping | 70 (35.0) | 38 (38.0) | 32 (32.0) | 0.37 [b] |
| Reading the results 15 minutes after adding the diluent | 88 (44.0) | 47 (47.0) | 41 (41.0) | 0.39 [b] |
| *No. of participants who made at least one error, n (%)* | 144 (72.0) | 74 (74.0) | 70 (70.0) | 0.53 [b] |
| *No. of participants who made at least one error (excluding timekeeping and reading results within stipulated time), n (%)* | 145 (72.5) | 74 (74.0) | 71 (71.0) | 0.64 [b] |
| **No. of participants with the following observed difficulty, n (%)** | | | | |
| Opening the package | 0 (0.0) | 0 (0.0) | 0 (0.0) | - |
| Pricking a finger | 22 (11.0) | 16 (16.0) | 6 (6.0) | 0.024 [b] |
| Obtaining a blood drop | 23 (11.5) | 13 (13.0) | 10 (10.0) | 0.51 [b] |
| Collecting the blood drop with the transfer device | 54 (27.0) | 26 (26.0) | 28 (28.0) | 0.75 [b] |
| Dispensing the diluent on the specimen well | 5 (2.5) | 3 (3.0) | 2 (2.0) | >0.95 [a] |
| Reading the results | 1 (0.5) | 1 (1.0) | 0 (0.0) | >0.95 [a] |
| *No. of participants with observed difficulties for at least one step, n (%)* | 75 (37.5) | 41 (41.0) | 34 (34.0) | 0.31 [b] |
| **No. of participants requiring assistance, n (%)** | | | | |
| Opening the package | 0 (0.0) | 0 (0.0) | 0 (0.0) | - |
| Obtaining a blood drop | 1 (0.5) | 1 (1.0) | 0 (0.0) | >0.95 [a] |
| Transferring the blood drop to the device | 10 (5.0) | 5 (5.0) | 5 (5.0) | >0.95 [b] |
| Dispensing the diluent onto the sample well | 4 (2.0) | 1 (1.0) | 3 (3.0) | 0.62 [a] |
| Reading the results | 0 (0.0) | 0 (0.0) | 0 (0.0) | - |
| *No. of participants requiring assistance for at least one step, n (%)* | 25 (12.5) | 15 (15.0) | 10 (10.0) | 0.29 [b] |

GP, general population; MSM, men who have sex with men.

[a] Fisher's exact test

[b] Pearson's chi-square test

without any difficulty and 87.5% completed the self-test procedure without any assistance. Compared with the general population, MSM were less likely to experience difficulties (6.0% vs. 16.0%; *p* = 0.024) and make errors (1.0% vs. 11.0%; *p* = 0.003) when pricking their fingers with a lancet (Table 3).

### Result interpretation after conducting HCVST

In this study, no positive result was detected by trained study staff re-reading the self-test nor by trained study stuff administering a professional test. For both the oral fluid-based and blood-based HCVST, there were high inter-reader concordances among the general population and MSM groups at 97.0% vs 99.0% and 99.0% vs 98.0% respectively (S1 Table). The Gwet's AC1 coefficients were 0.76 and 0.98 for the oral fluid-based HCVST among the general population and MSM, respectively, while they were 0.98 and 0.60 for the blood-based HCVST among the general population and MSM, respectively. Inconsistent result interpretation between readers was documented in four cases of oral fluid-based testing (three participants from the general population read their negative results as positive or invalid, while one MSM read their invalid result as negative) and in three cases of blood-based testing (one participant from the general population read their invalid result as negative, while two MSM read their negative results as positive or invalid).

The inter-operator concordances for the oral fluid-based HCVST were 91.0% and 98.0% among the general population and MSM, respectively; for the blood-based HCVST, the inter-operator concordances were 97.0% and 96.0% among the general population and MSM, respectively. The Gwet's AC1 coefficients were 0.89 and 0.99 for the oral fluid-based HCVST among the general population and MSM, respectively, while they were 0.99 and 0.99 for the blood-based HCVST among the general population and MSM, respectively. The results obtained from the professional-use tests conducted by trained users were different from those reported by the participants in eleven and seven cases of oral fluid- and blood-based tests, respectively. Among the eleven discordant cases involving oral fluid-based HCVST, one participant in the general population who interpreted their result as positive, was negative when retested by a trained user, while eight and two participants among the general population and MSM, respectively and who interpreted their results as invalid, were all negative on retesting. Among the seven discordant cases involving blood-based HCVST, one MSM participant, who interpreted their result as positive, was negative when retested by a trained user, while three participants in each of the general population and MSM groups and who interpreted their results as invalid, were all negative on retesting.

### Contrived result interpretation

Participants were generally able to read positive, negative, and invalid contrived results for both the oral fluid- and blood-based tests (Table 4). Clinically, all positive results (whether strong or weak) should be interpreted as positive. Accuracy of result interpretation was similar across both study populations and types of HCVST. Overall, majority of the contrived test results were interpreted correctly. Positive results with a clear test line and negative results were most accurately interpreted by the participants with over 97% of correct interpretations. Weak positives with a faint test line and invalid results with test line only posed some difficulties for interpretation. Contrived oral fluid HCVST devices displaying invalid test line-only results were read correctly by 62.5% of the participants, while similar blood-based HCVST devices were read correctly by 78.5% of the participants. However, weak positive results with a faint test line and a clear control were least accurately interpreted by the participants (51.3% for blood-based HCVST and 64% for oral fluid-based HCVST).

### Acceptability of HCVST

The acceptability of HCVST among the participants did not appreciably change after they had experienced both types of tests (Table 5). Although more than half (52.5%) of all participants were unaware of hepatitis C treatment and its availability in their communities, 97.0% of them

Table 4. Assessment of accuracy in the interpretation of test results based on contrived tests[#].

| Test result | Correct interpretation, n (%) | | | | | | | |
| --- | --- | --- | --- | --- | --- | --- | --- | --- |
| | Oral fluid-based test | | | | Blood-based test | | | |
| | Overall (n = 200) | GP (n = 100) | MSM (n = 100) | *p*-value | Overall (n = 200) | GP (n = 100) | MSM (n = 100) | *p*-value |
| Positive (with clear control and test lines) | 196 (98.0) | 98 (98.0) | 98 (98.0) | >0.95 [a] | 193 (97.5) | 98 (98.0) | 97 (97.0) | >0.95 [a] |
| Weak positive (with a clear control line and a faint test line) | 128 (64.0) | 60 (60.0) | 68 (68.0) | 0.24 [b] | 103 (51.3) | 50 (50.0) | 53 (53.0) | 0.67 [b] |
| Negative (with only a clear control line) | 194 (97.0) | 97 (97.0) | 97 (97.0) | >0.95 [a] | 195 (97.5) | 98 (98.0) | 97 (97.0) | >0.95 [a] |
| Invalid (without either control or test lines) | 191 (95.5) | 96 (96.0) | 95 (95.0) | >0.95 [a] | 192 (96.0) | 97 (97.0) | 95 (95.0) | 0.72 [a] |
| Invalid (with only a clear test line) | 125 (62.5) | 63 (63.0) | 62 (62.0) | 0.88 [b] | 157 (78.5) | 81 (81.0) | 76 (76.0) | 0.39 [b] |
| TOTAL | **834 (83.4)** | **414 (82.8)** | **420 (84.0)** | **0.61 [b]** | **840 (84.0)** | **424 (84.8)** | **418 (83.6)** | **0.60 [b]** |

GP, general population; MSM, men who have sex with men.

[#] Contrived tests were premade test cassettes provided by the manufacturers OraSure Technologies, United States of America and Premier Medical Corporation, India.

[a] Fisher's exact test

[b] Pearson's chi-square test.

expressed their willingness to use HCVST again and 98.5% would recommend it to their family and friends. The participants also expressed a preference to perform HCV self-testing in a primary care centre (33.5%), by themselves at home (20.0%), or by health staff in a primary care centre (20.0%). Just over half (51.5%) of all participants preferred oral fluid- over blood-based HCVST. The main reasons given for preferring the oral fluid-based test, as captured by an open-ended question, were its pain-free nature and ease of use. Nevertheless, 22.5% of participants did not express a preference. The two groups of participants did not differ in their views of HCVST, except that MSM were less willing to take the test kits home to their family and friends (76.0% vs. 88.0%; *p* = 0.045). The main reasons MSM gave for being unwilling to bring the test kits home to their family and friends, as captured by an open-ended question, were a lack of knowledge and confidence to explain the tests.

## Discussion

To the best of our knowledge, this study is the first one to report the usability and acceptability of oral fluid- and blood-based HCVST in two populations in a country. By targeting both the general population and MSM, this study has also provided insights into the usability of HCVST among individuals at different levels of risk for HCV infection in Malaysia. The world is currently witnessing an expanding epidemic of sexually transmitted HCV among MSM [29, 30]; this study demonstrates that HCVST is a highly acceptable testing modality among MSM in a religiously and culturally conservative society such as Malaysia and hence has a high potential to enhance screening coverage in this high-risk population. Using HCVST in addition and as a complement to HCV testing services in primary care centres also represents another milestone for Malaysia, which has a relatively low prevalence of HCV in the general population [17, 31] and is seeking to micro-eliminate HCV by targeting specific populations [19, 20, 32, 33].

Our findings showed that for both HCVST types, the most common mistakes made by participants in this study were not practicing good timekeeping and not reading the results within the stipulated time period (participants tended to read the results once the specimen moved

**Table 5. Acceptability of HCVST and additional information regarding awareness of HCV and its treatment.**

| Aspect | Overall (n = 200) | Subgroup | | |
|---|---|---|---|---|
| | | GP (n = 100) | MSM (n = 100) | *p*-value |
| **Willing to use HCVST again, n (%)** | 194 (97.0) | 98 (98.0) | 96 (96.0) | 0.41 [a] |
| **Willing to recommend the test to family and friends, n (%)** | 197 (98.5) | 99 (99.0) | 98 (98.0) | >0.95 [b] |
| **Willing to take the test kits to family and friends, n (%)** | 164 (82.0) | 88 (88.0) | 76 (76.0) [c] | 0.045 [a] |
| **Preferred HCV testing approach, n (%)** | | | | 0.53 [a] |
| By oneself in a primary care centre | 67 (33.5) | 29 (29.0) | 38 (38.0) | |
| By oneself at home | 40 (20.0) | 24 (24.0) | 16 (16.0) | |
| By health staff in a primary care centre | 40 (20.0) | 20 (20.0) | 20 (20.0) | |
| During a regular check-up in a health facility | 14 (7.0) | 6 (6.0) | 8 (8.0) | |
| During a screening campaign | 4 (2.0) | 3 (3.0) | 1 (1.0) | |
| No specific preference | 35 (35.0) | 18 (18.0) | 17 (17.0) | |
| **Preferred HCVST type, n (%)** | | | | 0.10 [a] |
| Oral fluid-based | 103 (51.5) [d] | 59 (59.0) | 44 (44.0) | |
| Blood-based | 52 (26.0) | 21 (21.0) | 31 (31.0) | |
| No specific preference | 45 (22.5) | 20 (20.0) | 25 (25.0) | |
| **Will contact the health facility if HCVST result is positive, n (%)** | 163 (81.5) | 82 (82.0) | 81 (81.0) | 0.86 [a] |
| **Will take a confirmatory test if HCVST result is positive, n (%)** | 68 (34.0) | 28 (28.0) | 40 (40.0) | 0.10 [a] |
| **Awareness about hepatitis C treatment, n (%)** | | | | 0.56 [a] |
| Aware that there is treatment and cure | 71 (35.5) | 32 (32.0) | 39 (39.0) | |
| Aware that there is treatment but unsure of cure | 24 (12.0) | 12 (12.0) | 12 (12.0) | |
| Unaware that there is treatment and cure | 105 (52.5) | 56 (56.0) | 49 (49.0) | |
| **Awareness about availability of hepatitis C treatment in one's community, n (%)** | | | | 0.62 [b] |
| Available | 122 (61.0) | 62 (62.0) | 60 (60.0) | |
| Available but not nearby | 6 (3.0) | 4 (4.0) | 2 (2.0) | |
| Not available or not sure | 72 (36.0) | 34 (34.0) | 38 (38.0) | |

GP, general population; HCVST, hepatitis C virus self-testing; MSM, men who have sex with men.

[a] Pearson's chi-square test

[b] Fisher's exact test

[c] Main reasons for the unwillingness among MSM to bring the test kits to their family and friends, as captured in an open-ended question, were a lack of knowledge and confidence to explain the tests

[d] Main reasons for preferring the oral fluid-based test, as captured in an open-ended question, were its pain-free nature and ease of use.

across the result window or once the control line appeared). These critical steps could affect the sensitivity of a HCVST, leading to invalid results or false negatives. In our study, we observed 2 out of the 3 invalid results reported by participants were read by them prematurely–just after the specimen moved across the result window–and both turned to be negative when re-read by the trained study staff. We could not assess the likelihood of false negative results as we had no HCV positive cases in our study population, however, reading results before stipulated time could cause false negative results and should this user error persist, it could be a major issue during scale-up of HCVST. The study was conducted before a massive roll out of rapid testing for COVID-19 and it is possible that participants in Malaysia at the time were not sensitized to the use of rapid diagnostic tests (RDTs) and might be unaware of the importance of timekeeping when using RDTs. These mistakes were not observed in Egypt or Vietnam where RDTs have been widely used for screening and diagnosis of infectious diseases [21, 22]. Henceforth the usability in Malaysia might gradually improve as the population

gets more familiar with the use of RDTs for self-testing. Such mistakes could also be attributed to unclear instructions in the IFU and might be prevented by further optimizing the IFU and providing supplementary material such as video instructions and flyers. The use of a smartphone application with timer function during HCVST may also guide users to perform these critical steps accurately. While pictorial instructions have shown great potential elsewhere to overcome cultural barriers in promoting self-testing [34], our findings highlight the importance of improving the pictorial instructions in relation to timekeeping and reading the results within the stipulated time period before any scale-up of HCVST. Additionally, participants might have overlooked to keep the time and read results within the stipulated time period as they could be in a hurry to complete the study procedures. Earlier on, they have spent some significant time in the primary care centres for standard of care before undergoing the study procedures and could have wanted to return to work or school as soon as they could. In the near future, when HCVST is made available outside this study and when individuals can conduct HCVST in their home or at a preferred location, these mistakes might be significantly reduced.

Another very frequent error observed during the blood-based HCVST was skipping the hand washing step recommended prior to sampling. However, this error was considered less critical as it is unlikely to lead to invalid or false results, as indicated by an invalid rate of 2.0–4.0% and an inter-operator agreement of more than 96.0%. A further challenge with the blood-based HCVST was the need to obtain and transfer a blood sample, as 27.0% of all participants made mistakes transferring the blood drop to the test device. With regards to additional help provided to participants while they performed the two types of HCVST, transferring the blood drop to the test device required the highest degree of assistance (5.0%). This observation is consistent with an earlier study that reported challenges with specimen collection for blood-based HIVST [35]. Again, it is possible that errors in blood-based HCVST procedures were made as participants in Malaysia were not yet sensitized to the use of RDTs and would reduce over time. Consistent to this assumption, we observed that MSM generally experienced fewer difficulties than the general population when pricking their fingers to obtain blood, likely due to their past exposure to RDTs administered by healthcare workers or community workers for HIV or HCV. Difficulties in obtaining blood and handling the test device could discourage new self-testers; therefore, product refinement and simplified testing procedures are necessary [36]. More guidance on specimen collection, such as video-based demonstration [37], is warranted, especially if blood-based HCVST were to be extended to the general population.

Consistent with findings among MSM in Vietnam and Egypt [21, 22], our study indicated high inter-reader concordances and reliability in the interpretation of results for both oral fluid- and blood-based HCVST. Furthermore, the ability of the participants to differentiate between clear "positive", "negative", and "invalid" test results was evidenced by their correct interpretation of contrived test results. We observed substantial difficulties in interpreting weak positive results with a faint test line indicating a need for additional guidance on how to interpret such results. Supportive materials such as manufacturer's IFU, video instructions and flyers would need to highlight and emphasize on how to interpret weak positive results; moreover, these materials should also provide information ensuring that if people with risk factors are not able to understand or interpret such weak positive results, they would have sufficient support to access facility-based testing by healthcare workers. Again, these difficulties may have arisen as participants in Malaysia were not yet sensitized to the use of RDTs and hence, such problems would reduce over time.

Also consistent with findings from Egypt and Vietnam [21, 22], the participants expressed a strong willingness to use HCVST again and to recommend it to their family and friends. They also expressed a clear preference for oral fluid- over blood-based testing, as did the general

population in Rwanda and PWID in Kyrgyzstan who had no actual experience of self-testing [38, 39]. In our study, the participants from both groups reported a preference for oral fluid-based HCVST that can be attributed to its ease of use and pain-free nature, which were also reasons for the high acceptability of oral fluid-based HIVST [11–14]. However, it is worth noting that for HIVST, studies have reported a greater preference for blood- over oral fluid-based self-tests [40, 41]. Only one-fifth of participants in our study preferred to perform HCVST by themselves at home. This may be due to lack of domestic privacy as observed with HIVST [12]. This may also imply that participants lacked confidence to perform the test alone by themselves at the time of the study, despite the written and pictorial instructions provided. A lack of knowledge and confidence to explain the self-test, as reported in responses to an open-ended question administered during this study, and the possible complication of punitive laws and discrimination against sexual minorities in Malaysia [42, 43], may also have contributed to the reduced willingness of MSM to take the test kits to family and friends. Together, these findings justify continuous efforts to optimize self-testing products, improve IFU, and establish a non-discriminatory hepatitis C care model, as recommended by the WHO [16].

The major limitation of this study lies in its focus on individuals who were willing to seek care, had access to health facilities and were enrolled based on their willingness to try HCVST. The structured questionnaires were limited in scope and could not fully capture participants' perceptions of the self-testing experience and acceptability of the self-tests. The usability and acceptability of HCVST among hard-to-reach populations, such as those who live in rural areas and are concerned about stigma, remain unclear. Furthermore, the absence of positive HCV test results among the participants limited the assessment of the inter-operator reliability of HCVST.

The MOH, in collaboration with the Malaysian AIDS Council and FIND, has recently initiated another study to address these limitations, mainly through the use of a web-based platform and online distribution to reach out to more potential self-testers [44]. While self-testers with a positive HCV result are expected to present themselves for further care, the MOH also recognizes the need to enhance public awareness of hepatitis C and the availability of curative treatment in health facilities across the country.

## Conclusion

This study demonstrated that both oral fluid- and blood-based HCVST were highly acceptable among both the general population and MSM in Malaysia. The general population and MSM showed comparable ability to conduct the tests and interpret the results. However, the frequencies of making critical errors related to timekeeping and reading results within stipulated time period were very high. Overall, this study suggests that both oral fluid- and blood-based HCVST could be introduced as an addition to existing HCV testing services for populations at different levels of risk for HCV infection in Malaysia, provided that sufficient guidance and clearer IFU (particularly highlighting the need for proper timekeeping and reading within the stipulated time) are in place.

## Supporting information

**S1 Fig. Contrived results for both the oral fluid- and blood-based tests.** [1]Positive (with clear control and test lines); [2]Weak positive (with a clear control line and a faint test line); [3]Negative (with only a clear control line); [4]Invalid (without either control or test lines); [5]Invalid (with only a clear test line).
(TIF)

**S1 Table. Assessment of inter-reader and inter-operator agreement in test result interpretation.**
(DOCX)

## Acknowledgments

The authors would like to thank the Government of Netherlands for the support; study participants for their involvement; MOH study teams for the implementation of this study; the following civil society organizations for their support in the recruitment of participants—PT Foundation and Intan Life Zone Welfare Society; OraSure Technologies, United States of America, and Premier Medical Corporation, India, for the in-kind contribution of research-use only HCV self-test kits for this study. The authors would also like to thank Cheryl Johnson, Mohammed Jamil, Niklas Luhmann and Philippa Easterbrook for their contributions in conceptualizing and developing protocols for the HCVST evaluation studies; and the Director-General of Health, Malaysia, for his support throughout the conduct of this study and permission to publish the findings.

Editorial support, under the direction of the authors and funded by FIND, was provided by Adam Bodley.

## Author Contributions

**Conceptualization:** Elena Ivanova Reipold, Emmanuel Fajardo, Sonjelle Shilton.

**Data curation:** Huan-Keat Chan, Xiaohui Sem.

**Formal analysis:** Huan-Keat Chan, Xiaohui Sem.

**Funding acquisition:** Elena Ivanova Reipold.

**Investigation:** Huan-Keat Chan, Xiaohui Sem, Sheela Bai A/P Pannir Selvam, Narul Aida Salleh, Abdul Hafiz Bin Mohamad Gani, Sonjelle Shilton, Muhammad Radzi Abu Hassan.

**Methodology:** Elena Ivanova Reipold, Emmanuel Fajardo, Sonjelle Shilton.

**Project administration:** Huan-Keat Chan, Xiaohui Sem, Elena Ivanova Reipold, Sheela Bai A/P Pannir Selvam, Narul Aida Salleh, Abdul Hafiz Bin Mohamad Gani, Emmanuel Fajardo, Sonjelle Shilton, Muhammad Radzi Abu Hassan.

**Supervision:** Huan-Keat Chan, Xiaohui Sem, Elena Ivanova Reipold, Sheela Bai A/P Pannir Selvam, Narul Aida Salleh, Abdul Hafiz Bin Mohamad Gani, Emmanuel Fajardo, Sonjelle Shilton, Muhammad Radzi Abu Hassan.

**Validation:** Huan-Keat Chan, Xiaohui Sem.

**Writing – original draft:** Huan-Keat Chan.

**Writing – review & editing:** Huan-Keat Chan, Xiaohui Sem, Elena Ivanova Reipold, Sheela Bai A/P Pannir Selvam, Narul Aida Salleh, Abdul Hafiz Bin Mohamad Gani, Emmanuel Fajardo, Sonjelle Shilton, Muhammad Radzi Abu Hassan.

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
