## [Decision Letter · Decision Letter 0]

18 May 2023

PGPH-D-23-00397

Usability and acceptability of oral fluid- and blood-based hepatitis C virus self-testing among the general population and men who have sex with men in Malaysia

Dear Dr. Sem,

Thank you for submitting your manuscript to PLOS Global Public Health. After careful consideration, we feel that it has merit but does not fully meet PLOS Global Public Health’s publication criteria as it currently stands. Therefore, we invite you to submit a revised version of the manuscript that addresses the points raised during the review process.so 

We look forward to receiving your revised manuscript.

Kind regards,

Andrew D. Kerkhoff

Academic Editor

Journal Requirements:

1. We do not publish any copyright or trademark symbols that usually accompany proprietary names, eg  ©, ®, ™  (e.g. next to drug or reagent names). Please remove all instances of trademark/copyright symbols throughout the text, including ® on page 7.

Additional Editor Comments (if provided):

Reviewers' comments:

Reviewer's Responses to Questions

**Comments to the Author**

1. Does this manuscript meet PLOS Global Public Health’s publication criteria? Is the manuscript technically sound, and do the data support the conclusions? The manuscript must describe methodologically and ethically rigorous research with conclusions that are appropriately drawn based on the data presented.

Reviewer #1: Yes

Reviewer #2: Partly

2. Has the statistical analysis been performed appropriately and rigorously?

Reviewer #1: Yes

Reviewer #2: I don't know

3. Have the authors made all data underlying the findings in their manuscript fully available (please refer to the Data Availability Statement at the start of the manuscript PDF file)?

Reviewer #1: Yes

Reviewer #2: Yes

4. Is the manuscript presented in an intelligible fashion and written in standard English?

Reviewer #1: Yes

Reviewer #2: Yes

5. Review Comments to the Author

Reviewer #1: Excellent study, which is very timely as we aim to scale up options for HCV testing and surveillance. Unlike for HIVST where the market was shaped mostly around Oral Based tests, this study placed equal precedence on Oral and Blood which for me is an important factor. HCVST has the potential to close the ever widening knowledge gap amongst those affected with HCV, a treatable disease, and studies such as this are vitally important.

Reviewer #2: This is an interesting and timely article on HCV self-testing usability among MSM and the general population in Malaysia. It will make an important contribution to the literature and the authors are commended on great work. I especially enjoyed the insightful comments about groups developing capability to perform RDTs with repeated use, and the thorough exploration of HCVST usability. I have some concerns about the reporting of the findings and outline major revisions which will strengthen this paper for publication.

Major

My most major concern is around how the figures for correct interpretation of positive results are reported. Currently the authors divide these up into ‘positive’ and ‘weak positive’ with only minimal discussion of the differences between these, and no discussion of the clinical implications of a weak positive. As all positive results (whether faint or strong) are indeed positive results and should be interpreted thus, it isn’t appropriate to only report these separately. Rather, all should be aggregated into one row and reported in the results. The authors may then also wish to report them separately in table 5. This is especially important given that participants often did not recognise ‘weak’ positive results (as high as 50% of the time for blood based tests in the general population).

A related point, I would suggest the authors expand with more detail on the implications of these issues with correctly interpreting tests results. Currently it received little attention despite being a critical issue to resolve before implementation.

The authors compare HCVST to HIVST a few places in the article. One critical point in this is missing which is that HCV antibody tests will remain positive after a person has cleared HCV, either through treatment or spontaneously during acute infection. That means that these tests are not necessarily as widely useful, especially in treatment experienced populations and key populations with repeated exposures. This issue also has really important implications for intervention design.

Minor

Line 51 – authors state 9.4 million HCV-infected individuals benefited from testing and treatment. Two issues here, 1) authors should use person centred language (e.g. people with HCV) and 2) mention if the 9.4 million is a global number (if it is?)

Line 60 – the systematic review referenced has been superseded by the following articles:

Witzel TC, Eshun-Wilson I, Jamil MS, Tilouche N, Figueroa C, Johnson CC, Reid D, Baggaley R, Siegfried N, Burns FM, Rodger AJ. Comparing the effects of HIV self-testing to standard HIV testing for key populations: a systematic review and meta-analysis. BMC medicine. 2020 Dec;18(1):1-3.

Eshun-Wilson I, Jamil MS, Witzel TC, Glidded DV, Johnson C, Le Trouneau N, Ford N, McGee K, Kemp C, Baral S, Schwartz S. A systematic review and network meta-analyses to assess the effectiveness of human immunodeficiency virus (HIV) self-testing distribution strategies. Clinical Infectious Diseases. 2021 Aug 15;73(4):e1018-28.

Jamil MS, Eshun-Wilson I, Witzel TC, Siegfried N, Figueroa C, Chitembo L, Msimanga-Radebe B, Pasha MS, Hatzold K, Corbett E, Barr-DiChiara M. Examining the effects of HIV self-testing compared to standard HIV testing services in the general population: A systematic review and meta-analysis. EClinicalMedicine. 2021 Aug 1;38:100991.

Line 64 – oddly phrased. Suggest starting this sentence with something about how it is critical to generate local evidence on acceptability, values and preferences to inform HCVST implementation.

Line 72 – explain what FIND is

Line 111 – worth mentioning whether the tests were inactivated (I believe they were not)

Line 124 – how were people assigned to groups

Line 167 – reflect somewhere on why such a high proportion of the group with living with HIV. Was one of the study sites an HIV treatment centre?

Line 327 – Language a bit confusing, worth clarifying

Line 366 – This issue could also be about domestic privacy, as is seen in a lot of settings with HIVST

6. PLOS authors have the option to publish the peer review history of their article (what does this mean?). If published, this will include your full peer review and any attached files.

**Do you want your identity to be public for this peer review?** For information about this choice, including consent withdrawal, please see our Privacy Policy.

Reviewer #1: **Yes: **Mohammed Majam

Reviewer #2: No

---

## [Decision Letter · Decision Letter 1]

17 Jul 2023

PGPH-D-23-00397R1

Usability and acceptability of oral fluid- and blood-based hepatitis C virus self-testing among the general population and men who have sex with men in Malaysia

Dear Dr. Sem,

Thank you for submitting your manuscript to PLOS Global Public Health. After careful consideration, we feel that it has merit but does not fully meet PLOS Global Public Health’s publication criteria as it currently stands. Therefore, we invite you to submit a revised version of the manuscript that addresses the points raised during the review process.

We look forward to receiving your revised manuscript.

Kind regards,

Andrew D. Kerkhoff

Academic Editor

Journal Requirements:

Additional Editor Comments (if provided):

Thank you for your revised submission. In addition to addressing the additional reviewer comments, please also address the minor, stylistic points below, which are largely focused on improving the readability of the manuscript.

Minor:

- Line 32: Would highlight in the abstract that weak positive results were difficult to read as this is an important finding. For example… “including that of contrived results, although there was difficulty interpreting weak positive results.

- Remove Sentence at lines 81-82 as it does not substantially add to the rationale for the study.

- Line 175: Add “the” before majority.

- Line 179: Add “compared to MSM” at the end of this sentence.

- Lines 198- 204: please separate each error by commas rather than a semi-colon.

- Line 224: given the large amount of text that follows this subheader, consider adding specificity to this subheader and including an additional subheader at line 257 for improved clarity and readability.

- Line 263: Add “a” - Weak positives with (a) faint test line.

- Line 340: add “the” - … was skipping (the) hand washing step…

- Line 363: remove “to” - … indicating a need for…

- Line 373: it appears that a reference is missing.

Table 1. Please round any p-values greater than 0.05 to two decimal points only. Further do not bold “significant values.” Also change “latest” to “Most recent” for HCV and HIV test results for improved clarity.

Table 2. Please round any p-values greater than 0.05 to two decimal points only. For consistency, add a row that corresponds to the “number of participants who made at least one error in pre-testing-steps.” For readability, it would be helpful to indent each of the rows that correspond to the above sub-header. This would help to more clearly separate each usability testing subsection. It would also help to not bold the cumulative measures and to also indent them under the corresponding header. Perhaps these could be italicized instead.

Table 3. Similar comments as Table 2 with regard to formatting considerations.

Table 4. Given the large number of tables included, I would consider making this a supplementary table given the key aspects are well summarized in the results section.

Table 5. Please round any p-values greater than 0.05 to two decimal points only. Consider adding a brief footer that gives a quick summary of what a ‘contrived” test is for readers less familiar with usability testing and this term.

Table 6. Please round any p-values greater than 0.05 to two decimal points only.

Reviewers' comments:

Reviewer's Responses to Questions

**Comments to the Author**

1. If the authors have adequately addressed your comments raised in a previous round of review and you feel that this manuscript is now acceptable for publication, you may indicate that here to bypass the “Comments to the Author” section, enter your conflict of interest statement in the “Confidential to Editor” section, and submit your "Accept" recommendation.

Reviewer #2: (No Response)

2. Does this manuscript meet PLOS Global Public Health’s publication criteria? Is the manuscript technically sound, and do the data support the conclusions? The manuscript must describe methodologically and ethically rigorous research with conclusions that are appropriately drawn based on the data presented.

Reviewer #2: No

3. Has the statistical analysis been performed appropriately and rigorously?

Reviewer #2: I don't know

4. Have the authors made all data underlying the findings in their manuscript fully available (please refer to the Data Availability Statement at the start of the manuscript PDF file)?

Reviewer #2: Yes

5. Is the manuscript presented in an intelligible fashion and written in standard English?

Reviewer #2: Yes

6. Review Comments to the Author

Reviewer #2: Thank you for the opportunity to review this revised manuscript. The majority of my comments have been addressed and I feel this article is clearer. I remain enthusiastic about this publication and think it will make an important contribution to the literature.

I maintain substantial concern around the discussion of weak positives, and it feels as though important findings are being downplayed. The difficulty in interpreting weak positives is a critical finding which problematises HCVST and must be address for successful implementation. Given that half of participants were not able to correctly interpret faint positive blood-based HCVSTs (and nearly half oral fluid ones), the proposed revisions are not objective (or reflective of reality).

The authors state: “Participants were generally able to read positive, weak positive, negative, and invalid contrived results for both the oral fluid- and blood-based tests (Table 5). Clinically, all positive results (whether strong or weak) should be interpreted as positive.”; and Line 363 - “We observed some difficulties in interpreting weak positive results with a faint test line indicating to a need for additional guidance on how to interpret such results. Again, these difficulties may have arisen as participants in Malaysia were not yet sensitized to the use of RDTs and hence, such problems would reduce over time.”

The participants were indeed not generally able to read weak positive blood-based results and the difficulties should not be characterised as ‘some’ but rather as ‘substantial’. It’s helpful to highlight the need for further advice, what other intervention components could be included?

Finally, the changes also need to be reflected in the abstract so that those skimming the article are aware from the outset.

7. PLOS authors have the option to publish the peer review history of their article (what does this mean?). If published, this will include your full peer review and any attached files.

**Do you want your identity to be public for this peer review?** For information about this choice, including consent withdrawal, please see our Privacy Policy.

Reviewer #2: No

---

## [Editor Report · Decision Letter 2]

21 Nov 2023

Usability and acceptability of oral fluid- and blood-based hepatitis C virus self-testing among the general population and men who have sex with men in Malaysia

PGPH-D-23-00397R2

Dear Dr. Ivanova Reipold,

We are pleased to inform you that your manuscript 'Usability and acceptability of oral fluid- and blood-based hepatitis C virus self-testing among the general population and men who have sex with men in Malaysia' has been provisionally accepted for publication in PLOS Global Public Health.

Best regards,

Max Carlos Ramírez-Soto, BSc, MPH, FRSPH, MACE

Academic Editor

No comments